# Beyond Pixels: Mining Compressed Domain Artifacts for Efficient AI-Generated Video Detection

**Anran Zhu** [1]  **Zhengli Shi** [*1]  **Chende Zheng** [*1]  **Chenhao Lin** [#1]  **Zhengyu Zhao** [1]  **Le Yang** [1]  **Chong Zhang** [1]  **Shuai Liu** [2]  **Chao Shen** [1]

## Abstract

With the rapid advancement of high-fidelity video generation models, robust AI-generated video (AIGV) detection has become increasingly needed. While most AIGV detection methods operate in the decoded pixel domain, we observe that detection in the pixel domain inevitably entangles task-irrelevant semantic information, leading to substantial semantic redundancy and extensive redundant computation, while overlooking free-to-use signals in compressed bitstreams. In particular, motion vectors and residuals directly encode temporal and spatial generative artifacts but remain largely underexplored. To address these issues, we propose a unified framework for **S**patio-**T**emporal **RE**sidual and **A**rtifact **M**ining, namely **STREAM**, which enables AIGV detection directly from compressed bitstreams. **STREAM** leverages I-frames, motion vectors, and residual errors to capture spatiotemporal artifacts that are typically smoothed out by decompression filters. In particular, we design a lightweight network with a motion-guided alignment module and a gated fusion mechanism, enabling adaptive fusion of spatial artifacts and nonlinear temporal dynamics. Extensive experimental results demonstrate that **STREAM** achieves SOTA performance with an mAP of 0.965, with 2.5× faster inference than previous SOTA baselines.

## 1. Introduction

AI-generated videos are increasingly blurring the boundary between reality and fiction. Recently, the successive emergence of general-purpose video generation frame-

---
[*]Equal contribution  [#] Corresponding author  [1]School of Cyber Science and Engineering, Xi'an Jiaotong University [2]School of Software Engineering, Xi'an Jiaotong University. Correspondence to: Chenhao Lin <linchenhao@xjtu.edu.cn>.

*Proceedings of the $43^{rd}$ International Conference on Machine Learning*, Seoul, South Korea. PMLR 306, 2026. Copyright 2026 by the author(s).

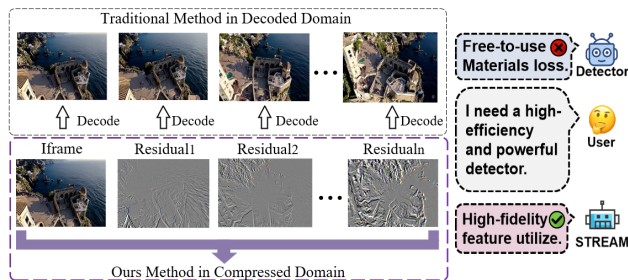

*Figure 1.* Comparison between pixel-domain video detection and the proposed compressed-domain framework. Conventional methods necessitate fully decoding MPEG-4/H.264 encoded videos into RGB frames before processing detection tasks with complex backbones. In contrast, our method operates directly on compressed representations, including I-frames, motion vectors, and residual errors, enabling efficient and artifact-preserving detection.

works such as Sora, Veo(Google DeepMind, 2025), and CogVideoX(Yang et al., 2024) has brought AI-generated videos into the public spotlight. Correspondingly, a growing body of research has investigated AI-generated content detection, with representative approaches including DeCoF(Ma et al., 2024), DeMamba(Chen et al., 2024), and NPR(Tan et al., 2024). In contrast to image forgery detection tasks, video forgery detection involves temporal inconsistencies and motion-related artifacts that cannot be reliably captured by image-based detectors, resulting in poor transferability of the latter to video scenarios.

Existing AI-generated video detection methods operate in the decoded pixel domain, aiming to extract generated artifacts from RGB frames.

Although this paradigm is the most straightforward, it suffers from two fundamental limitations: (1) pixel-domain detectors incur prohibitive computational costs, as full video decoding and dense frame-level processing introduce substantial redundancy; (2) Decoding inevitably suppresses or smooths fine-grained generated artifacts, while completely discarding valuable information that already exists in the video compressed domain.

Notably, we observe that the compressed bitstreams contain several free-to-use components, including I-frames, Motion

Vectors (MVs), and Residuals, that are generated during standard video encoding. These elements inherently encode rich forensic signals: Residuals preserve spatial inconsistencies similar to the artifacts exploited by Neighboring Pixel Relationships (NPR) (Tan et al., 2024), while MVs serve as an efficient proxy for optical flow, capturing temporal irregularities introduced during synthesis (Ji et al., 2024). Importantly, obtaining these cues in pixel-domain methods typically requires expensive computation, whereas they are directly available in the compressed domain without full decoding.

Some prior work has explored the utilization of compressed bitstreams for video vision tasks such as object detection and action recognition, primarily to improve inference efficiency. However, for these tasks, operating in the compressed domain often entails a trade-off between speed and accuracy, as semantic understanding relies heavily on pixel-level fidelity. In contrast, AI-generated video detection hinges on identifying low-level artifacts—signals that are often distinctively preserved in the compressed domain yet obscured during reconstruction.

This distinction highlights a critical inefficiency in current methods: fully decoding videos for detection introduces three major drawbacks: (1) excessive and unnecessary computation; (2) complete neglect of rich compression-domain information; and (3) degradation of subtle generative artifacts during the decoding process. Despite this potential, to the best of our knowledge, no prior work has explored AI-generated video forgery detection in the video compressed domain to date.

To bridge this gap, we propose **STREAM** (**S**patio-**T**emporal **RE**sidual and **A**rtifact **M**ining), a novel framework that performs AI-generated video detection directly from compressed bitstreams. Instead of relying on full decoding, STREAM leverages complementary compressed-domain representations by processing I-frames, Motion Vectors, and Residuals through a lightweight multi-branch architecture. To effectively aggregate these heterogeneous modalities, we introduce a Motion-Guided Alignment Module to calibrate spatial misalignment and a Gated Fusion Mechanism to adaptively integrate static spatial artifacts with dynamic temporal inconsistencies. This design minimizes computational overhead while preserving fine-grained signals typically lost during full decoding. Extensive experiments demonstrate that our approach achieves state-of-the-art accuracy across diverse generation models and datasets, while remaining robust to common post-processing operations and delivering substantially faster inference than pixel-domain baselines. In summary, this paper makes three key contributions:

- We are the first to systematically investigate generative artifacts in the video compressed domain, revealing

that widely used pixel-domain detectors predominantly learn redundant task-irrelevant semantic content, rather than intrinsic forgery cues.

- We propose STREAM, an efficient AI-generated video detection framework that operates directly on compressed domains. By explicitly mining spatiotemporal generative artifacts from I-frames, motion vectors, and residuals, STREAM achieves strong generalization across diverse video generation models.

- Extensive experiments on 5 public benchmarks (including 44 test sets, covering 28 different video generation models) show that STREAM consistently outperforms the previous SOTA methods by 2.2%~29.3% mAP, and achieves a 2.5× faster time cost than the previous SOTA pixel-domain detectors derived from the pixel domain. Moreover, STREAM remains robust under real-world post-processing operations.

## 2. Related Work

### 2.1. AI for Video Generation

Recent general video generation algorithms are predominantly built upon spatiotemporal diffusion architectures. These models, including Sora, Veo (Google DeepMind, 2025), and Keling (Kuaishou, 2025), model spatiotemporal dependencies by simulating the reverse noise diffusion process, aligning textual semantics with dynamic physical priors to generate high-fidelity content. Approaches like CogVideoX (Yang et al., 2024) and SVD (Blattmann et al., 2023) further refine this by employing multi-stage diffusion strategies or extending pre-trained image models with temporal attention. However, these generation methods inevitably undergo upsampling processes, which unavoidably introduce distinct generation artifacts that serve as critical cues for detection (Tan et al., 2024).

### 2.2. AI-Generated Video Detection

Mainstream detection methods primarily operate on fully decoded RGB frames, branching into two main paradigms: passive forensics based on spatiotemporal artifacts and active reasoning based on consistency. In the former category, Zhang (Chen et al., 2024) and Tan (Tan et al., 2024) analyze frequency domain anomalies and local neighboring pixel relationships (NPR) to identify microscopic generation traces, while D3 (Zheng et al., 2025) characterizes temporal artifacts via inter-frame optical flow variations. Conversely, consistency-based methods verify high-level semantic or physical coherence. For instance, DeCoF (Ma et al., 2024) and DEFAKE (Sha et al., 2023) leverage CLIP to detect semantic mutations, DeMamba (Chen et al., 2024) employs the Mamba architecture for efficient spatiotemporal modeling, and NSG (Zhang et al., 2025) introduces a

probability flow conservation metric to quantify violations of physical laws. Additionally, AIGVDet (Bai et al., 2024) utilizes a dual-stream network to independently assess spatial and temporal consistency. In terms of datasets, VidProm (Wang & Yang, 2024) introduces a dataset that contains AI-generated videos along with their corresponding textual descriptions (prompts), which also provides a foundation for detection methods based on spatiotemporal consistency.

Despite these advancements, pixel-domain approaches face significant inherent limitations. First, decoding compressed videos (typically H.264/MPEG-4) into RGB frames inevitably invokes loop filters that attenuate subtle generation artifacts, diminishing detection cues. Second, modeling complex spatiotemporal irregularities from high-dimensional pixel data incurs prohibitive computational costs and latency, hindering real-time applications. Most critically, these methods neglect the rich, accessible information embedded directly in the compressed bitstream, such as residual errors and motion vectors, leading to redundant computations and the loss of raw forensic signals.

### 2.3. Video Compression in Vision Tasks

Compressed-domain representations have been widely adopted in video understanding tasks to improve efficiency. CoViAR (Wu et al., 2018) decomposes videos into I-frames, motion vectors (MVs), and residuals for fast action recognition, while CVOS (Wang et al., 2019) and MMNet (Xu & Yao, 2022) further exploit these "free-to-use" cues for efficient feature propagation and fusion. Typically, these methods perform heavy semantic modeling on I-frames and rely on lightweight networks for residuals, using motion vectors to propagate features across time.

However, for semantic recognition tasks, compressed-domain methods often trade accuracy for speed. In contrast, forgery detection fundamentally differs from semantic understanding: it focuses on capturing subtle generative artifacts rather than restoring high-level semantics. Decoding videos to recover semantic-rich RGB frames is therefore unnecessary, and even detrimental, for artifact detection. To the best of our knowledge, no prior work has systematically explored AI-generated video detection in the compressed domain. Our work bridges this gap by explicitly leveraging and fusing compressed-domain modalities to characterize spatiotemporal generative artifacts, enabling efficient, accurate, and robust AI-generated video detection.

## 3. Methodology

### 3.1. Motivation

Most existing AI-generated video detection methods operate directly on decoded RGB frames. However, this pixel-domain analysis faces a fundamental limitation: the video decoding process acts as a lossy filter. To optimize perceptual quality, codecs (e.g., H.264) apply post-processing operations—such as deblocking and interpolation—that inevitably smooth out the subtle high-frequency artifacts introduced by generative models. Consequently, forensic traces that are evident in the raw signal become attenuated in the decoded pixel domain, making detection significantly more challenging and computationally expensive.

To recover these traces, recent approaches attempt to explicitly extract residual information from decoded frames using learned denoisers like DnCNN (Zheng et al., 2024). While effective at suppressing semantic content, these methods rely on a *secondary* extraction process applied after the decoding damage is already done. In contrast, the compressed domain offers direct access to low-level coding features—specifically residuals and motion vectors, which are generated during the encoding process. These features represent the primitive signal traces of the video prior to reconstruction, effectively bypassing the artifact-suppressing filters of the decoder.

Consequently, the compressed domain offers a more faithful representation for forensic analysis through two primary mechanisms. On the spatial front, prediction residuals retain the original high-frequency generative artifacts, avoiding the attenuation caused by decoding filters that limit pixel-based counterparts. On the temporal front, the motion vectors natively available in the bitstream can directly reveal the non-physical motion irregularities and jitter often exhibited by synthetic videos, avoiding the potential errors of optical flow estimation. These properties allow for a detection framework that is both physically grounded and robust to the masking effects of video reconstruction.

**Artifacts in Residuals.** Modern video codecs apply a series of post-processing operations during decoding, such as deblocking and interpolation filters, to improve perceptual quality. Although visually beneficial, these operations inevitably suppress subtle high-frequency artifacts introduced by generative models, making forensic analysis in the decoded pixel domain more challenging.

To mitigate this issue, several pixel-domain methods explicitly extract residual representations. For example, prior works employ learned denoisers, such as DnCNN (Corvi et al., 2023; Zheng et al., 2024), to remove semantic content and isolate fine-grained high-frequency residual signals, which have been proven effective for detecting AI-generated images and videos. However, such residuals are obtained through a *secondary operation* applied after decoding, and therefore inherit the inherent information loss introduced by the decoding process itself entirely.

To quantitatively examine the difference between these residual representations, we analyze their spectral characteristics

using the radial power spectrum. The analysis is conducted on samples drawn from multiple public AI-generated video benchmarks, including VidProM, GenVideo, GVF, Video-Phy, and AIGVDET-D. For each dataset, we randomly select 100 videos per class and uniformly sample 8 frames from each individual video, resulting in a diverse set of frames for comprehensive spectral analysis altogether.

Given the Fourier transform $F(u, v)$ of a signal $I(x, y)$, the one-dimensional radial power spectrum $P_{1D}(r)$ is computed by integrating the power spectrum $P(r, \theta)$ over the angular dimension:

$$P_{1D}(r) = \frac{1}{N_r} \sum_{\theta} P(r, \theta), \qquad (1)$$

where $r$ denotes the spatial frequency radius, $\theta$ represents the angular component, and $N_r$ is the number of frequency components at radius $r$.

As illustrated in Figure 2 (a), we compare three core representations:(i) original compressed-domain residuals,(ii) pixel-domain residuals extracted via a DnCNN denoiser,and (iii) standard decoded RGB frames. The empirical results show that the real–fake divergence in the mid-to-high frequency bands is significantly larger for compressed-domain residuals than for DnCNN-based pixel residuals. This observation indicates that secondary residual extraction, while effective in suppressing semantics, cannot recover the high-frequency forensic artifacts that are already attenuated during standard decoding altogether.

**Physical Inconsistency in Motion Vectors.** Generative video models often fail to strictly adhere to real-world physical constraints (Ma et al., 2024; Bai et al., 2024; Zheng et al., 2025), resulting in microscopic temporal jitter and a lack of inertia. To exploit this, prior works such as AIGVDet (Bai et al., 2024) utilize optical flow to characterize these anomalies. However, we observe that motion vectors in the compressed stream natively encode this motion field, allowing us to detect such non-physical movements without the heavy computational burden of optical flow estimation.

We quantify this phenomenon using a Local Spatial Consistency metric (using the same data samples as the previous residual analysis). Based on the hypothesis that real physical motion (e.g., rigid bodies) exhibits local smoothness, we measure the directional consistency of motion vectors:

$$S_t = \frac{1}{|M|} \sum_{p \in M} \text{CircDist}(\theta_p, \overline{\theta}_{\mathcal{N}(p)}) \qquad (2)$$

where $M$ represents the set of all macroblocks in the current frame, $p$ is the index of the current macroblock, $\theta_p$ is its motion direction, and $\overline{\theta}_{\mathcal{N}(p)}$ denotes the average direction of its 8-neighboring macroblocks. The function $\text{CircDist}(\cdot)$

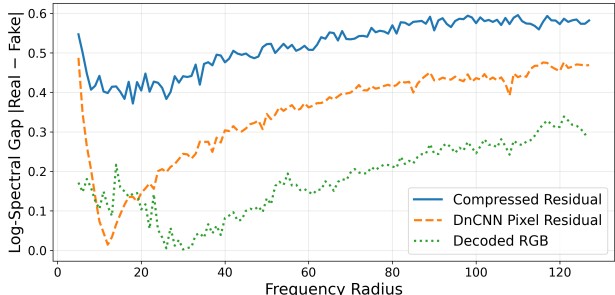

(a) Radial spectrum difference comparison (spatial artifacts)

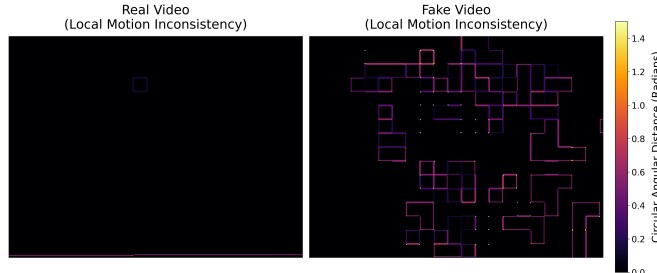

(b) Motion vector inconsistency comparison (temporal artifacts)

*Figure 2.* Spatio-temporal artifact analysis. (a) Radial spectrum divergence shows that compressed-domain residuals preserve more discriminative spatial artifacts than pixel-domain counterparts. (b) Motion vector inconsistency maps reveal pronounced temporal noise patterns in forged videos.

computes the circular distance between two angles, defined as the shortest distance on the unit circle to properly account for angular periodicity:

$$\text{CircDist}(\alpha, \beta) = \min(|\alpha - \beta|, 2\pi - |\alpha - \beta|) \qquad (3)$$

As visualized in Figure 2 (b), our analysis shows that real videos exhibit smooth and coherent motion fields (low $S_t$), whereas deepfake videos manifest high-frequency, star-like noise patterns in their consistency heatmaps, indicating a breakdown of local physical motion constraints. These observations motivate our proposed architecture, which explicitly exploits the uncorrupted residuals and physical motion inconsistencies present in the compressed domain.

### 3.2. Method

To overcome the computational redundancy and information loss inherent in pixel-domain detection, we propose the **S**patio-**T**emporal **RE**sidual and **A**rtifact **M**ining (**STREAM**) framework. Instead of reconstructing RGB frames, STREAM operates directly on the native MPEG-4 compressed bitstream, leveraging three computationally efficient modalities: I-frames, motion vectors (MVs), and prediction residuals.

As illustrated in Figure 3, STREAM consists of two core

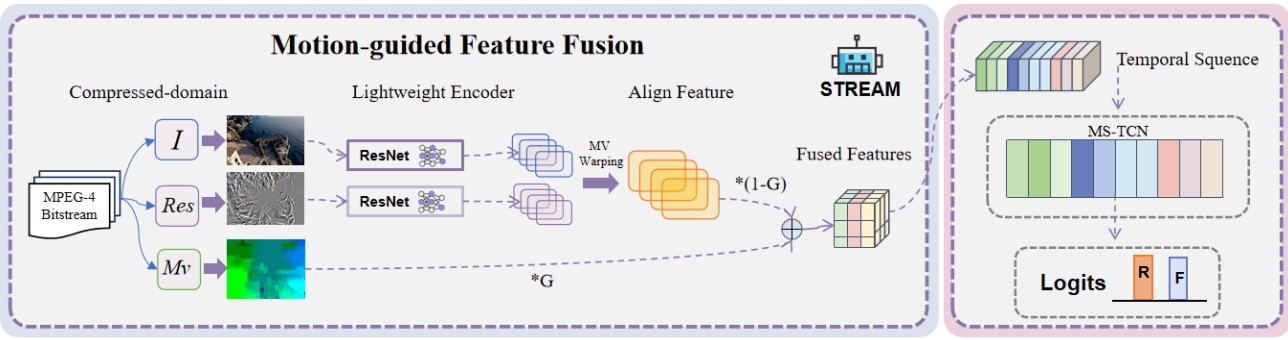

*Figure 3.* The overall architecture of the proposed STREAM framework. It leverages native compressed domain features to detect generative artifacts through motion-guided fusion and multi-scale temporal modeling.

stages: (1) **Motion-Guided Feature Fusion**, which aligns and adaptively integrates spatial texture and compression artifacts; and (2) **Temporal Inconsistency Mining**, which captures temporal anomalies across multiple time scales for video-level forgery detection.

### 3.2.1. MOTION-GUIDED FEATURE FUSION

The compressed bitstream provides heterogeneous signals with distinct physical meanings: I-frames encode static texture information, motion vectors describe inter-frame displacement, and residuals capture prediction errors. To effectively exploit their complementary properties, we design a motion-guided fusion mechanism that performs spatial alignment and adaptive artifact selection.

**Stream-Specific Encoding.** Each modality is processed by an independent lightweight encoder to learn modality-specific representations without mutual interference. Given an input video clip, the encoders produce feature maps:

$$F_I, F_{Res}^{(t)} \in \mathbb{R}^{C \times H \times W} \qquad (4)$$

where $t$ indexes the temporal dimension.

**Motion-Based Spatial Alignment.** Due to inter-frame motion, features extracted from I-frames are spatially misaligned with those of predictive frames. Leveraging the geometric displacement encoded in motion vectors, we align the I-frame features to each temporal step via feature-level warping:

$$F_{Align}^{(t)} = \text{Warp}(F_I, MV^{(t)}), \qquad (5)$$

where $\text{Warp}(\cdot)$ denotes a differentiable motion compensation operation. This alignment corrects spatial offsets without requiring explicit optical flow estimation. $MV^{(t)}$ represents the motion vector of the current frame.

**Adaptive Artifact Gating.** Generative artifacts are typically sparse and localized, often manifesting in high-frequency residual regions. To selectively emphasize these

regions, we introduce a pixel-level gating mechanism that adaptively balances residual and texture features. Specifically, a gate map is computed as:

$$G^{(t)} = \sigma \left( \text{Conv}_{1 \times 1} \left( \left[ F_{Res}^{(t)}, F_{Align}^{(t)} \right] \right) \right), \qquad (6)$$

where $[\cdot]$ denotes channel-wise concatenation and $\sigma$ is the Sigmoid function. The gate $G^{(t)} \in [0, 1]^{1 \times H \times W}$ quantifies the relative importance of residual artifacts versus aligned texture context.

**Gated Feature Integration.** The final fused feature for each frame is obtained via a soft selection mechanism:

$$F_{Fused}^{(t)} = G^{(t)} \cdot F_{Res}^{(t)} + \left( 1 - G^{(t)} \right) \cdot F_{Align}^{(t)}. \qquad (7)$$

This adaptive fusion highlights forensic artifacts in manipulated regions while preserving semantic texture in visually consistent areas.

### 3.2.2. TEMPORAL INCONSISTENCY MINING

While spatial artifacts provide local evidence, deepfake videos often exhibit temporal inconsistencies such as flickering, jitter, or physically implausible dynamics. To capture such anomalies, we aggregate frame-level representations into a temporally coherent video-level descriptor.

**Global Feature Aggregation.** For each frame $t$, the fused spatial feature map $F_{Fused}^{(t)}$ is compressed into a compact vector via Global Average Pooling (GAP):

$$v_t = \frac{1}{H \times W} \sum_{i=1}^{H} \sum_{j=1}^{W} F_{Fused}^{(t)}(i, j), \qquad (8)$$

where $v_t \in \mathbb{R}^C$. This produces a temporal feature sequence $V = \{v_1, v_2, \ldots, v_T\}$.

**Multi-Scale Temporal Dynamics Modeling.** The sequence $V$ serves as the input to a Multi-Scale Temporal

Convolutional Network (MS-TCN), where the initial hidden state is defined as:

$$h_t^{(0)} = v_t. \qquad (9)$$

Each temporal convolution layer employs dilated 1D convolutions to capture dependencies at different time scales:

$$h_t^{(l)} = \text{ReLU}\left(\sum_{k=0}^{K-1} W_k^{(l)} \cdot h_{t-k \cdot d_l}^{(l-1)} + b^{(l)}\right), \qquad (10)$$

where $l$ indexes the layer, $d_l$ is the dilation factor, and $K$ is the kernel size. By progressively increasing $d_l$, the network effectively models both short-term temporal jitter and long-range incoherence.

**Video-Level Classification.** After $L$ temporal layers, the resulting features $\{h_t^{(L)}\}_{t=1}^{T}$ are aggregated via temporal average pooling to form a global video representation:

$$z_{global} = \frac{1}{T}\sum_{t=1}^{T} h_t^{(L)}. \qquad (11)$$

A fully connected layer followed by Softmax produces the final prediction:

$$\hat{y} = \text{Softmax}(W_{fc} \cdot z_{global} + b_{fc}), \qquad (12)$$

where $\hat{y}$ denotes the probability of the input video being pristine or AI-generated. The entire framework is trained end-to-end using binary cross-entropy loss applied at the video level.

# 4. Experiments

## 4.1. Experimental Setup

**Implementation Details.** All runtime measurements are conducted on a single NVIDIA GeForce RTX 4090 GPU. The reported inference time includes bitstream parsing, motion vector and residual extraction, and tensorization, excluding disk I/O operations. We use a batch size of 4 for inference and 64 for training. Our framework also supports flexible architectural configurations. Unless otherwise specified, the default setting in the main experiments employs ResNet152 as the encoder for the I-frame branch and ResNet18 for the residual branch.

**Evaluation Metrics.** To comprehensively evaluate the performance of our proposed framework, we utilize two core metrics. (i) We employ the Mean Average Precision (mAP) as the primary indicator. It is calculated by determining the area under the Precision-Recall (P-R) curve across varying confidence thresholds, providing a holistic assessment of the model's ability to discriminate between positive samples (AI-generated videos) and negative samples

(real videos). (ii) We quantify the computational speed by measuring the total time required to process 1,000 standardized video clips. A lower time value corresponds to faster processing speed, highlighting the inherent efficiency advantages of our compressed-domain approach compared to pixel-domain baselines.

**Datasets.** We conduct model training on a small subset of the VidProM(Wang & Yang, 2024) dataset, which comprises 10,000 video samples generated by SVD(Blattmann et al., 2023), VC2(IJCAI Proceedings) and PIKA(Labs, 2024). To evaluate generalization, we conducted testing on a diverse set of benchmarks and datasets, including Gen-Video(Chen et al., 2024), GVF(Ma et al., 2024), Video-Phy(Bansal et al., 2024), AIGVDet-T2V-Dataset (AIGVDet-D)(Bai et al., 2024).

**Baselines.** We compare our model against several state-of-the-art baseline methods: D3 (ICCV'25) (Zheng et al., 2025), DeMamba (Chen et al., 2024), NPR (CVPR'24) (Tan et al., 2024), STIL (MM'21) (Gu et al., 2021), TALL (ICCV'23) (Xu et al., 2023), and AIGVDet (PRCV'24) (Bai et al., 2024).

## 4.2. Generalization Experiments

We evaluate the proposed framework on two key aspects: detection accuracy across multiple datasets and inference efficiency (Table 1). Both evaluations allow for fair comparison against state-of-the-art baselines.

**Detection Performance.** Table 1 reports the mAP of STREAM and baselines across five datasets (VidProM, Gen-Video, GVF, VideoPhy, and AIGVDet-D). Our compressed-domain method consistently outperforms pixel-domain baselines, particularly on unseen generative models, demonstrating strong generalization. This advantage stems from the compressed domain preserving fine-grained artifacts that are often smoothed out during standard pixel-domain decoding. Quantitatively, STREAM achieves the highest average mAP of 0.965 across all datasets, surpassing the second-best baseline, D3 (0.949), by 0.016. Specifically, on VidProM, STREAM reaches an mAP of 0.999 (+0.023 over D3). Even on GVF, where the performance gap is narrowest, STREAM maintains a clear lead with 0.924 mAP compared to D3's 0.895. This consistent superiority verifies the intrinsic robustness of our framework altogether.

**Inference Efficiency.** As shown in Table 1, STREAM demonstrates significant speed advantages by bypassing full decoding and utilizing a lightweight architecture. Processing 1,000 standardized clips takes only 16 seconds per 1,000 videos, making STREAM the fastest among all compared methods. This represents a 60% efficiency improvement

*Table 1.* Comparison on detection results (average precision) and inference efficiency (seconds per 1,000 videos). **Bold** represents the best and underline represents the second.

| MODEL | VIDPROM | GENVIDEO | GVF | VIDEOPHY | AIGVDET-D | mAP | TIME (S) |
|---|---|---|---|---|---|---|---|
| D3 (ICCV'25) | 0.976 | 0.961 | 0.895 | 0.958 | 0.924 | 0.943 | 40 |
| DEMAMBA (ARXIV'24) | 0.917 | 0.881 | 0.841 | 0.911 | 0.863 | 0.883 | 91 |
| NPR (CVPR'24) | 0.777 | 0.792 | 0.723 | 0.823 | 0.752 | 0.773 | 188 |
| STIL (MM'21) | 0.867 | 0.813 | 0.765 | 0.867 | 0.757 | 0.814 | 80 |
| TALL (ICCV'23) | 0.756 | 0.686 | 0.612 | 0.674 | 0.632 | 0.672 | 103 |
| AIGVDET (PRCV'24) | 0.964 | 0.918 | 0.882 | 0.942 | 0.890 | 0.919 | 74 |
| OURS (STREAM) | **0.999** | **0.978** | **0.924** | **0.974** | **0.950** | **0.965** | **16** |

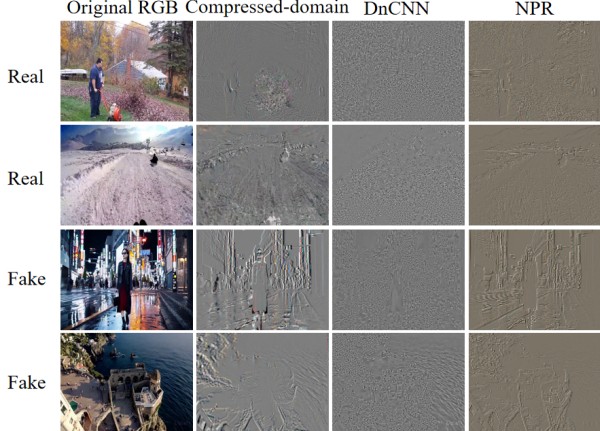

*Figure 4.* Comparison of compressed-domain artifacts in real and fake videos with those derived from pixel-domain post-processing (e.g., DnCNN/NPR).

over the second-fastest D3 (40 seconds per 1,000 videos) and an 11.75× speedup over the slowest baseline, NPR (188 seconds per 1,000 videos). Crucially, STREAM breaks the traditional accuracy-efficiency trade-off, delivering state-of-the-art detection performance while meeting real-time deployment requirements.

### 4.3. Visualization

To further validate the effectiveness of compressed-domain features, we conduct a qualitative analysis by visualizing compressed-domain residuals, residuals obtained from pixel-domain preprocessing via DnCNN, and upsampling artifacts from NPR. As can be seen from Figure 4, compressed-domain residuals exhibit remarkably superior discriminability, a conclusion further corroborated by their similarity to the upsampling artifacts of NPR. More importantly, Figure 2 in section 3 verifies that compressed-domain residuals possess better discriminative capability than residuals acquired via DnCNN preprocessing. Table 6 in subsection 4.4 further demonstrates that compressed-domain residuals achieve notably higher detection performance than NPR upsampling artifacts. Both NPR and DnCNN rely on pixel-domain processing, which inevitably introduces additional computational overhead and degrades feature purity due to seman-

*Table 2.* Ablation experiments on performance and inference time.

| MODULE | VIDEOPROM | GENVIDEO | VIDEOPHY | GVF | AIGVDET-D | TIME (S) |
|---|---|---|---|---|---|---|
| I-FRAME | 0.999 | 0.988 | 0.932 | 0.885 | 0.984 | 6.9 |
| MV | 0.966 | 0.943 | 0.943 | 0.991 | 0.912 | 3.4 |
| RESIDUAL | 0.996 | 0.996 | 0.915 | 0.941 | 0.944 | 3.6 |
| I+R+M | 0.999 | 0.978 | 0.974 | 0.950 | 0.965 | 16.0 |

tic interference. Furthermore, the residuals derived from DnCNN usually contain redundant noise signals and fail to restore authentic tampering traces. Such inherent drawbacks of the two processing methods also intuitively illustrate that compressed-domain residual features can achieve higher detection efficiency.

### 4.4. Ablation Study

**Impact on Inference Efficiency.** To further analyse the impact of each component on framework efficiency, we conduct an ablation study on inference times for different module combinations (Table 2). We employed the metric 'time (s/1,000 videos)' to measure the computational cost of processing 1,000 standardised video frames, where lower values indicate higher inference efficiency.

As shown in Table 2, single-module experiments reveal significant computational overhead differences: the lightweight motion vector (MV) and residual modules achieved exceptionally fast inference speeds, taking 3.4 seconds and 3.6 seconds per 1,000 videos, respectively, whereas the I-frame module took nearly twice as long (6.9 seconds). This result indicates that compared with compressed domain representations (motion vectors and residuals), processing inter-frame features starting from the pixel domain requires more computational resources. After integrating all three modules, our STREAM achieves a total inference time of 16 seconds when processing 1,000 video frames.

**Impact on Detection Accuracy.** To comprehensively validate the complementary effects of different compression domain components, we conduct module-level ablation studies. Table 2 presents the detection accuracy (mAP) of individual modules (I-frame, Motion Vector, and Residual) and their fusion scheme (I+R+M) across four representative

*Table 3.* Detection performance (mAP) of STREAM with different ResNet backbones on AI-generated video datasets.

| Method | sora | Crafter | Gen2 | HotShot | Lavie | ModelScope | MoonValley | MorphStudio | Show_1 | WildScrape | mAP |
|---|---|---|---|---|---|---|---|---|---|---|---|
| STREAM(ResNet-18) | 0.8824 | 0.9787 | 0.9888 | 0.7749 | 0.9282 | 0.9080 | 0.9946 | 0.9510 | 0.9208 | 0.8216 | 0.9149 |
| STREAM(ResNet-152) | 0.9131 | 0.9737 | 0.9885 | 0.8554 | 0.8857 | 0.8970 | 0.9900 | 0.9402 | 0.8938 | 0.8667 | 0.9204 |

*Table 4.* Computational complexity and parameter quantity comparisons (30 FPS). GFLOPs denotes giga floating-point operations, and Params denotes the number of parameters in millions.

| MODEL | GFLOPs/s (30 FPS) | PARAMS (M) |
|---|---|---|
| STREAM(RESNET-18) | 164.1 | 33.54 |
| STREAM(RESNET-152) | 457.5 | 80.51 |
| RAFT | 21105.6 | 46.99 |
| AIGVDET | 975.2+21105.6 | 46.99+5.26 |

AI-generated video datasets.

As illustrated, when STREAM integrates all three core modules (I+R+M), the framework achieves stable high mAP ($\geq$0.95) across all tested datasets: it retains the inherent high accuracy of individual modules while effectively compensating for their weaknesses (e.g., elevating the I-frame module's GVF score from 0.885 to 0.95). This demonstrates that the strategic fusion of spatial (inter-frame) features with compressed-domain temporal (MV) and residual (R) features creates complementary representations, enabling the framework to adapt to diverse generation mechanisms and achieve robust universal detection.

**Computational Complexity Analysis.** To quantitatively evaluate the efficiency of the proposed STREAM framework, we further analyze its computational complexity, measured in GFLOPs, and model size, measured by the number of parameters (M). As shown in Table 3, STREAM employs two backbone variants that achieve highly comparable detection performance (mAP: 0.9204 vs. 0.9149), demonstrating the effectiveness of the proposed lightweight design.

Detailed complexity comparisons are reported in Table 4. Under the lightweight configuration, STREAM with ResNet-18 only uses 33.54M parameters and 164.1 GFLOPs. The STREAM with ResNet-152 backbone contains 80.51M parameters and costs 457.5 GFLOPs at 30 FPS. Despite the performance improvement, both variants maintain substantially lower effective computational complexity than state-of-the-art approaches such as AIGVDet and RAFT. This efficiency gain mainly stems from the fact that STREAM operates directly on sparse compressed-domain representations, including motion vectors and residuals, rather than dense RGB frames, thereby significantly reducing the computational burden of feature extraction and backbone processing.

**Impact of Gaussian blurring.** We employ Gaussian blurring to explicitly simulate the post-compression effects

of information propagation on video in typical practical real-world scenarios, with (Strong/Medium/Weak) blur kernel sizes of ((3,3)/(5,5)/(7,7)), respectively. The results are shown in Table 5. It can be observed that within the STREAM detection method, the I-frame (I) and residual (R) components are notably highly susceptible to noise or post-compression effects, whereas the spatio-temporal information from the Motion Vector (MV) demonstrates remarkably greater inherent robustness, effectively compensating for pure spatial information's vulnerability to common transient transmission disturbances.

### 4.5. Robustness Experiments

**Robustness to multi-round re-encoding Experiments.** To directly verify that decoding-induced loop filtering attenuates forensic signals and that compressed-domain representations outperform pixel-domain ones, we conduct multi-round re-encoding experiments. The experimental results are presented in Table 6, where *Filter_cnt* denotes the number of times loop filtering is applied during repeated decoding and re-encoding processes.

As shown in Table 6, detection accuracy of all methods gradually declines as loop filtering rounds increase. This verifies that decoding loop filtering weakens subtle generative artifacts and degrades forensic features. Across all re-encoding settings, STREAM consistently outperforms AIGVDet and NPR with the slightest performance drop. This demonstrates its superior robustness to compression distortion over pixel-domain detectors, and further validates our core claim: loop filtering suppresses generative artifacts in decoded frames, while compressed-domain features better preserve forensic cues for detection.

To thoroughly address the concern about whether re-encoding preprocessing introduces additional filtering that may affect forensic signal preservation, we clearly clarify that our re-encoding preprocessing does not introduce extra filtering. All videos are transcoded under a default no-filter setting, ensuring that this preprocessing step does not weaken our claim about the preservation of generative artifacts in the compressed domain.

**Robustness to Realistic Post-Processing Experiments.** As shown in Table 7, STREAM achieves an average mAP of 0.9587 under the baseline setting. It exhibits strong robustness against resolution changes and H.265 re-encoding, with only negligible performance drops. H.264 re-encoding

*Table 5.* Evaluation results of different video generation models under iframe/mv/residual dimensions.

| FEATURE | STRENGTH | COGVIDEOX | COGVIDEOX-5B | ZEROSCOPE | DREAM_MACHINE | GEN2 | LAVIE | OPENSORA | PIKA | SVD | VC2 | mAP |
|---|---|---|---|---|---|---|---|---|---|---|---|---|
| IFRAME | ORIGINAL | 0.784 | 0.824 | 0.959 | 0.935 | 0.981 | 0.965 | 0.930 | 0.980 | 0.959 | 0.994 | 0.931 |
| IFRAME | WEAK | 0.836 | 0.912 | 0.930 | 0.971 | 0.987 | 0.890 | 0.936 | 0.993 | 0.954 | 0.994 | 0.940 +0.009 |
| IFRAME | MEDIUM | 0.743 | 0.842 | 0.826 | 0.924 | 0.968 | 0.919 | 0.878 | 0.967 | 0.942 | 0.983 | 0.899 -0.032 |
| IFRAME | STRONG | 0.702 | 0.778 | 0.750 | 0.924 | 0.955 | 0.930 | 0.855 | 0.967 | 0.924 | 0.971 | 0.876 -0.056 |
| MV | ORIGINAL | 0.883 | 0.906 | 0.866 | 0.929 | 0.917 | 0.913 | 0.983 | 0.987 | 0.936 | 0.954 | 0.937 |
| MV | WEAK | 0.883 | 0.942 | 0.942 | 0.965 | 0.962 | 0.965 | 0.988 | 0.947 | 0.977 | 0.977 | 0.955 +0.017 |
| MV | MEDIUM | 0.854 | 0.918 | 0.913 | 0.935 | 0.949 | 0.948 | 0.983 | 0.974 | 0.959 | 0.959 | 0.940 +0.003 |
| MV | STRONG | 0.830 | 0.901 | 0.866 | 0.924 | 0.910 | 0.924 | 0.983 | 0.974 | 0.942 | 0.988 | 0.924 -0.013 |
| RESIDUAL | ORIGINAL | 0.896 | 0.866 | 0.971 | 0.988 | 0.994 | 0.895 | 0.924 | 0.987 | 0.994 | 0.994 | 0.941 |
| RESIDUAL | WEAK | 0.832 | 0.897 | 0.924 | 0.888 | 0.981 | 0.744 | 0.779 | 0.974 | 0.994 | 0.988 | 0.900 -0.041 |
| RESIDUAL | MEDIUM | 0.814 | 0.914 | 0.872 | 0.913 | 0.994 | 0.645 | 0.651 | 0.974 | 0.983 | 0.994 | 0.875 -0.066 |
| RESIDUAL | STRONG | 0.879 | 0.856 | 0.802 | 0.935 | 0.994 | 0.488 | 0.640 | 0.947 | 0.959 | 0.924 | 0.842 -0.099 |
| I+R+M | ORIGINAL | 0.902 | 0.961 | 0.952 | 1.000 | 1.000 | 0.961 | 0.971 | 1.000 | 1.000 | 1.000 | 0.975 |
| I+R+M | WEAK | 0.883 | 0.959 | 0.901 | 1.000 | 0.994 | 0.942 | 0.930 | 1.000 | 0.994 | 1.000 | 0.960 -0.014 |
| I+R+M | MEDIUM | 0.854 | 0.912 | 0.907 | 0.988 | 1.000 | 0.936 | 0.971 | 1.000 | 1.000 | 1.000 | 0.957 -0.018 |
| I+R+M | STRONG | 0.825 | 0.825 | 0.942 | 0.988 | 0.994 | 0.948 | 0.971 | 1.000 | 1.000 | 1.000 | 0.949 -0.026 |

*Table 6.* Performance comparison of different methods under multi-round re-encoding (loop filter application count varies).

| FILTER$_{CNT}$ | 0 | 1 | 2 | 3 | 4 | 5 | 6 |
|---|---|---|---|---|---|---|---|
| AIGVDET | 0.915 | 0.828 | 0.708 | 0.670 | 0.643 | 0.644 | 0.594 |
| NPR | 0.928 | 0.867 | 0.765 | 0.709 | 0.644 | 0.614 | 0.602 |
| STREAM (OURS) | 0.976 | 0.950 | 0.874 | 0.788 | 0.750 | 0.712 | 0.697 |

*Table 7.* Performance under post-processing operations.

| Post-Processing | Setting | Videophy | GenVideo | GVF | Avg. mAP | Δ mAP |
|---|---|---|---|---|---|---|
| None (baseline) | — | 0.974 | 0.978 | 0.924 | 0.959 | — |
| Re-encoding | H.264, QP=28 | 0.979 | 0.957 | 0.902 | 0.946 | -0.0130 |
| Re-encoding | H.264, QP=35 | 0.936 | 0.913 | 0.858 | 0.902 | -0.0570 |
| Resolution change | 480p | 0.985 | 0.965 | 0.911 | 0.954 | -0.0050 |
| Resolution change | 720p | 0.989 | 0.969 | 0.915 | 0.958 | -0.0010 |
| Watermark | 50% opacity | 0.861 | 0.837 | 0.783 | 0.827 | -0.1320 |
| Re-encoding | H.265 | 0.986 | 0.976 | 0.911 | 0.958 | -0.0010 |

leads to slight accuracy degradation, especially under high quantization parameter (QP) values. Larger QP values indicate heavier compression intensity. Notably, the 50% opacity watermark causes the most significant performance drop, which indicates that the model is sensitive to occlusion interference. Overall, our method maintains stable detection.

## 5. Limitation

Although STREAM demonstrates strong detection performance and high efficiency, it still has several limitations in real-world deployment. In particular, its performance may degrade on manually spliced or tampered videos. This is because STREAM relies on the continuity of compressed-domain motion representations. Specifically, for P-frames referencing previous non-I frames, the framework accumulates and converts inter-frame motion vectors into displacement offsets aligned with I-frames before feature extraction. Manual clipping or splicing operations disrupt this temporal motion continuity, resulting in inconsistent motion propagation patterns that may lead to misclassification.

In addition, STREAM currently has limited adaptability to different video compression standards. It does not natively support newer codecs such as HEVC and AV1. Improving robustness against complex editing operations and extending support to more compression formats remain important directions for future work.

## 6. Conclusion

Our work argues that the core research paradigm for AI-generated video detection should focus on generative artifact detection rather than semantic recognition. Systematic analysis thoroughly shows that residuals and motion vectors can effectively characterize spatiotemporal artifacts of AI-generated videos while avoiding redundant semantic computation and pixel-domain semantic interference. Based on the above, we fuse three core modalities (I-frames, residuals, motion vectors) in the compressed domain and propose the STREAM detection framework—the first efficient method for full compressed-domain AI-generated video detection. Eliminating the need to decode video bitstreams into full-pixel frames and the lightweight STREAM design enable fast AI-generated video detection. Extensive experiments convincingly verify that STREAM achieves state-of-the-art AI-generated video detection performance across generative models and strong robustness against diverse realistic post-processing operations.

The STREAM framework proposed herein is fully implemented in the compressed domain, providing a new research entry point for AI-generated video detection. Future research directions include unified feature representation across compression standards, i.e., exploring a unified feature representation method for multiple video compression standards (e.g., H.265, AVI) to further improve the detection framework's universality.

## Acknowledgements

This work was supported by the Fundamental and Interdisciplinary Disciplines Breakthrough Plan of the Ministry of Education of China (No. JYB2025XDXM114), the National Natural Science Foundation of China (Nos. T2341003, 62521002, U2441240, U24B20185, 62376210, 62402377, 62536002), and the Shaanxi Provincial Key Research and Development Program (No. 2025ZG-JBGS-008). Thanks to the New Cornerstone Science Foundation and the Xplorer Prize.

## Impact Statement

This paper presents work whose goal is to advance the field of machine learning. There are many potential societal consequences of our work, none of which we feel must be specifically highlighted here.

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

To further fully validate the overall performance, generalization ability and practical robustness of the proposed STREAM framework, we carry out a series of supplementary experiments. We compare our method with the latest state-of-the-art NSG-VD algorithm, use accuracy metrics to cross-check detection results, specify detailed experimental data settings, and investigate model stability under extreme bandwidth constraints.

## A. NSG-VD vs STREAM

As shown in Table 8, we compare NSG-VD (NeurIPS'25) in our experimental setup. STREAM achieves better performance and higher efficiency than NSG-VD. Specifically, STREAM outperforms NSG-VD in average accuracy (0.9204 vs. 0.9146) while running nearly 9× faster (39 s vs. 360 s per 1,000 videos), highlighting STREAM's superior efficiency without sacrificing detection performance. The results are as follows:

*Table 8.* Comparison with NSG-VD on different AI-generated video datasets. STREAM achieves better overall accuracy while being nearly 9× faster.

| Method | sora | Crafter | Gen2 | HotShot | Lavie | ModelScope | MoonValley | MorphStudio | Show_1 | WildScrape | Average | Time(s/1000 video) |
|---|---|---|---|---|---|---|---|---|---|---|---|---|
| NSG-VD | 0.8837 | 0.9583 | 0.8833 | 0.9167 | 0.9417 | 0.8167 | 0.9667 | 0.9833 | 0.9083 | 0.8875 | 0.9146 | 360 |
| **Ours (STREAM)** | **0.9131** | **0.9737** | **0.9885** | 0.8554 | 0.8857 | **0.8970** | **0.9900** | 0.9402 | 0.8938 | 0.8667 | **0.9204** | **39** |

## B. The accuracy (ACC) metrics

As shown in Table 9, STREAM consistently achieves the best performance across all datasets and metrics, further confirming its superiority over existing baselines under both mAP and ACC evaluation criteria. We will add the ACC results in the revised manuscript.

*Table 9.* Accuracy (ACC) comparison across different datasets. STREAM consistently achieves the best performance on all benchmarks.

| Model | VidProM | GenVideo | GVF | VideoPhy | AIGVDet-D | Average Accuracy |
|---|---|---|---|---|---|---|
| D3 (ICCV'25) | 0.921 | 0.908 | 0.872 | 0.915 | 0.892 | 0.902 |
| DeMamba (arXiv'24) | 0.883 | 0.856 | 0.831 | 0.879 | 0.841 | 0.858 |
| NPR (CVPR'24) | 0.814 | 0.825 | 0.786 | 0.842 | 0.801 | 0.814 |
| STIL (MM'21) | 0.862 | 0.827 | 0.803 | 0.862 | 0.804 | 0.832 |
| TALL (ICCV'23) | 0.801 | 0.768 | 0.732 | 0.761 | 0.742 | 0.761 |
| AIGVDet (PRCV'24) | 0.915 | 0.887 | 0.864 | 0.903 | 0.871 | 0.888 |
| **Ours (STREAM)** | **0.948** | **0.932** | **0.897** | **0.941** | **0.918** | **0.927** |

## C. Data Setting

The real videos used for testing were sourced from the real-world collection in GenVideo, while the synthetic content was derived from a wide variety of popular generative models. The training and test sets contain videos encoded in H.264, MPEG-4, as well as videos in .gif format. The video data used in our experiments adopts the following encodings and formats:

- **H.264 encoding**: Used by all real video samples, as well as videos generated by GVF (GEN2, keling, T2V, Pika, show1, sora, Veo), VideoPhy (Cogvideox-1, CogVideox-5B, dream_machine, Gen2, LaVIE, OpenSora, Pika, SVD, vc2, zeroscope), VideoProM (Cogvideo, OpenSora, Pika, st2v, t2vc, vc2), and GenVideo (DynamicCrafter, Latte, OpenSora, SENIE, SD).

- **MPEG-4 encoding**: Used for videos generated by GVF (ModelScope, zeroscope) and VideoProM (ms_video).

- **GIF format**: Used for videos generated by GenVideo (i2vgen, SD).

To ensure a controlled and fair evaluation, all videos in both the training and test sets are uniformly re-encoded using the same standard MPEG-4 profile.

# D. Impact of extreme bandwidth constraints

Meanwhile, to evaluate model robustness under extreme bandwidth constraints, we conduct stress tests by transcoding videos to 300 kbps, a significant compression ratio compared to the original source content. Experimental results are shown in Table 10. Under the original bit rate, each feature and the fusion framework's overall detection mAP shows clear obvious gradient differences: STREAM fusion framework (I+R+M) achieves the best performance with an mAP of 0.975.

*Table 10.* Evaluation results of different video generation models under different representations and bit rates.

| FEATURE | BITE RATE | COGVIDEOX | COGVIDEOX-5B | ZEROSCOPE | DREAM_MACHINE | GEN2 | LaVIE | OpenSora | PIKA | SVD | VC2 | MAP |
|---|---|---|---|---|---|---|---|---|---|---|---|---|
| IFRAME | ORIGINAL | 0.784 | 0.824 | 0.959 | 0.935 | 0.981 | 0.965 | 0.930 | 0.980 | 0.959 | 0.994 | 0.931 |
| IFRAME | 300 KBPS | 0.678 | 0.678 | 0.849 | 0.882 | 0.949 | 0.802 | 0.657 | 0.954 | 0.930 | 0.948 | 0.861-0.071 |
| MV | ORIGINAL | 0.883 | 0.906 | 0.866 | 0.929 | 0.917 | 0.913 | 0.983 | 0.987 | 0.936 | 0.954 | 0.937 |
| MV | 300 KBPS | 0.866 | 0.866 | 0.843 | 0.918 | 0.885 | 0.913 | 0.988 | 0.942 | 0.948 | 0.954 | 0.921-0.016 |
| RESIDUAL | ORIGINAL | 0.896 | 0.866 | 0.971 | 0.988 | 0.994 | 0.895 | 0.924 | 0.971 | 0.994 | 0.981 | 0.941 |
| RESIDUAL | 300 KBPS | 0.885 | 0.897 | 0.907 | 0.947 | 0.994 | 0.905 | 0.738 | 0.984 | 0.988 | 0.942 | 0.914-0.027 |
| I+R+M | ORIGINAL | 0.902 | 0.961 | 0.952 | 1.000 | 1.000 | 0.961 | 0.971 | 1.000 | 1.000 | 1.000 | 0.975 |
| I+R+M | 300 KBPS | 0.784 | 0.754 | 0.919 | 0.971 | 0.981 | 0.820 | 0.866 | 0.987 | 0.994 | 0.988 | 0.905-0.070 |

When the video bit rate is reduced to 300 kbps, all features and the fusion framework's overall mAP decreases mildly slightly, indicating STREAM has strong robustness in extreme bit rate reduction scenarios. The decline range of each feature varies. Experiments show MV and residual features consistently provide STREAM reliable anti-interference ability against low-bit-rate compression, endowing the fusion framework with such capability. Though the fusion framework has a relatively large decline range, it still maintains high detection performance.

