# OpenReview forum: "Beyond Pixels: Mining Compressed Domain Artifacts for Efficient AI-Generated Video Detection"
_ICML.cc/2026/Conference — ICML 2026 regular_

### Official Review · Reviewer_3t5F · 2026-03-05

**Soundness:** 2
**Presentation:** 2
**Significance:** 3
**Originality:** 2
**Overall Recommendation:** 3
**Confidence:** 3

**Summary:**

This paper argues that artifacts in AI-generated videos become attenuated during the video decoding process. To address this issue, the authors propose detecting such artifacts using compressed-domain signals available before full decoding, namely I-frames, motion vectors, and residuals. They observe that I-frames capture spatial patterns, motion vectors reflect motion consistency, and residuals contain high-frequency artifacts that can serve as discriminative features between real and fake videos. Based on this observation, the method aligns spatial features using motion vectors and then combines the features through a gated linear fusion mechanism (Motion-Guided Feature Fusion). The fused representations are further processed along the temporal dimension through temporal feature modeling. By doing so, the approach removes the computational overhead of existing methods such as optical flow estimation and full video decoding, achieving improved efficiency while leveraging discriminative cues for strong detection performance.

**Compliance With Llm Reviewing Policy:**

Affirmed.

**Final Justification:**

Core concerns on novelty, justification of claims, and lack of clear empirical or theoretical support remain unresolved, so my evaluation remains unchanged.

**Key Questions For Authors:**

(1) The paper attributes the detection capability primarily to generative artifacts preserved in compressed-domain signals. Could the authors provide additional evidence that the model indeed relies on such artifacts rather than generic compression noise or dataset-specific correlations?

(2) The manuscript claims that decoding operations suppress generative artifacts. Could the authors clarify which specific decoding processes (e.g., deblocking, interpolation, inverse transforms) contribute to this effect and how these operations attenuate the relevant signals?

(3) Is there a new theoretical perspective on the role these components play in AIGV detection, and can this perspective be formalized (e.g., through a signal-level formulation or equations)?

**Limitations:**

Considering that the paper was submitted to the Social Aspects (Safety) track, it is recommended that the authors add a discussion section clearly outlining the methodological limitations of the proposed approach and the potential impacts that may arise from these limitations.

**Strengths And Weaknesses:**

**Strengths**
- The paper proposes an immediately deployable pipeline by leveraging a clear problem formulation and readily available compressed-domain features.
- The approach achieves substantial improvements in inference time by eliminating computationally expensive operations such as optical flow estimation and full video decoding.
- Choice of features is reasonably supported by empirical observations that highlight clear and interpretable feature characteristics.

**Weaknesses**
- Clarify the rationale for categorizing this work under Social Aspects (Safety). The current manuscript primarily focuses on the technical aspects of video detection and does not directly discuss societal impact or security-related issues.
- Main components (I-frame, motion vector, and residuals) were already introduced in earlier approaches such as CoViAR. The current work primarily interprets and observes these components from the perspective of AIGV detection, but it does not present a fundamentally new perspective.
- The core stages of the proposed framework appear to rely on largely standard components. Specifically, the Motion-Guided Feature Fusion module resembles a gated fusion mechanism with pixel-wise weighting, while the Temporal Inconsistency Mining module can be interpreted as temporal sequence modeling based on temporal convolutions. Although these design choices are reasonable and effective, the methodological novelty of the architecture appears somewhat limited.
- Several of the paper’s key arguments appear to rely largely on intuition or empirical observations rather than clearly established justification. In particular, the manuscript suggests that important signals in the compressed bitstream are suppressed during video decoding, yet it remains unclear which specific decoding operations lead to this effect or how these operations attenuate the relevant signals.
- Similarly, the paper attributes the discriminative power of residual representations to high-frequency artifacts, but it is not clearly demonstrated why such high-frequency components necessarily correspond to generative artifacts rather than generic noise or compression residuals.
- Moreover, although the framework is motivated by the goal of capturing generative artifacts, the training objective does not explicitly enforce artifact-related learning, which raises some uncertainty about whether the model indeed learns artifact-based cues or instead exploits other correlated patterns in the data.

Writing
- In Section 3.1 Artifacts in Residuals, space (“ ”) are missing in the second and third paragraphs, and also in the fifth paragraph.
- In Section 3.1 Artifacts in Residuals, the datasets cited in the third paragraph appear to be mentioned for the first time; the corresponding citations should be provided properly.
- Clearly define the variables ($x$, $y$, $u$, $v$).
- The green text in the tables is difficult to read; it is recommended to change it to a more visible color.

---

> ### Author Rebuttal · Authors · 2026-03-31
>
> ## **Response W1:**
> We will include a discussion on the broader societal implications of AIGV detection, covering misinformation, media integrity, and security-critical content authentication.
>
> ---
>
> ## **Response W2:**
> While CoViAR uses I-frames, MVs, and residuals for **semantic efficiency**, we are the first to reinterpret them as **carriers of spatiotemporal generative artifacts** for forensic detection，a fundamentally different perspective. The key insight is pixel-domain decoding (e.g., deblocking) actively suppresses high-frequency forensic signals, whereas compressed-domain features preserve them, making STREAM not only faster but inherently more faithful for detection.
>
> In response to the relevant question from **Reviewer ZA52 (Q3)**, we provide experimental evidence demonstrating that compressed-domain representations preserve forensic signals that pixel-domain counterparts lose.
>
> ---
>
> ## **Response W3:**
> STREAM’s novelty lies not in the individual building blocks, but in how they are instantiated. Our gate signal is derived from **motion vectors** with explicit geometric semantics, enabling $F^{(t)}_{MV}$ to serve dual roles simultaneously: (1) a
> **geometric displacement field** for I-frame feature warping (Eq. 4), and (2) a **forensic feature** capturing physically implausible motion patterns for gated fusion (Eq. 5–6). No prior gated fusion method exploits this dualityit is uniquely motivated by the structure of compressed bitstreams. We will clarify this in the revision.
>
> ---
>
> ## **Response W4 & Q2:**
> We clarify that the primary factor is the **deblocking filter** in H.264/AVC decoding, which acts as a **low-pass filter**. Prior work shows that generative artifacts introduced by upsampling layers in generative models are predominantly **high-frequency** (e.g., AutoGAN, WIFS'2019 and NPR, CVPR'24). As a result, deblocking directly attenuates these components, reducing forensic discriminability in the pixel domain.
>
> ---
>
> ## **Response W5:**
> We note that the correspondence between **generative model upsampling artifacts and high-frequency signals** is well-established in prior work.Our contribution is not to re-establish this fact, but to demonstrate that **compressed-domain residuals preserve these high-frequency generative artifacts more faithfully** than pixel-domain counterparts, as evidenced by our radial power spectrum analysis (Fig. 2a).
>
>
> **AutoGAN (WIFS'19)** analyzes frequency-domain fingerprints introduced by GAN upsampling operations. **NPR (CVPR'24)** demonstrates that neighboring pixel relationships reflect upsampling artifacts across diverse generative architectures. **FID (NeurIPS'24)** further characterizes the spectral properties of generated content. These studies consistently show that generative models introduce characteristic high-frequency patterns.
>
> ---
>
> ## **Response W6 & Q1:**
> We acknowledge that interpretability is inherently difficult to rigorously establish. Nevertheless, we provide complementary evidence through comparative results, OOD evaluation, and visualization analyses. We refer to the main results (**Tab. 1**), visualization (**Fig. 4**), feature space analysis (**Tab. R1**, response to **Reviewer gscv**) and the Generalization and Scaling Analysis (**Tab. R2**, response to **Reviewer gscv**).
>
> ---
>
> ## **Response Q3:**
> Our method reveals AI-generated video artifacts from a novel perspective. It introduces compressed-domain analysis and confirms that such artifacts are more richly captured in compressed-domain representations. Specifically, residuals encode block-level prediction errors that directly reflect spatial high-frequency inconsistencies introduced by generative models, and motion vectors expose temporal incoherence across frames, signals that pixel-domain methods do not explicitly model.
> This finding receives conceptual support from NPR (CVPR'24). Our residuals share a parallel with NPR's intra-frame differencing in suppressing low-frequency content while preserving generative artifacts, and our motion vectors extend this principle to the temporal domain. Together, this suggests that compressed-domain features are intrinsically aligned with the forensic signals targeted by artifact-based detectors, while offering a richer and more efficient alternative to pixel-domain analysis.
> We acknowledge that a rigorous signal-level formalization of this connection remains open and will include this as future work.
>
> ---
>
> ## **Response Limitations:**
> We appreciate the suggestion and will add a Limitations section in the revision:
> Because our method relies on continuous MV encoding, artificially spliced videos can disrupt MV continuity and might cause misclassifications. These vulnerabilities pose potential safety and societal risks, echoing the broader implications emphasized by this track.
>
> ---
>
> ## **Response Writing:**
> We will revise all writing and formatting issues in the revision.

---

> > ### Author Rebuttal · Reviewer_3t5F · 2026-04-01
> >
> > I appreciate that the rebuttal clarifies some technical aspects of the paper, and I acknowledge the practical appeal of the proposed pipeline in terms of efficiency and its use of compressed-domain signals for AIGV detection. However, my core concerns remain unresolved. First, the rationale for submitting this work to the safety track is still not sufficiently justified. Second, although the authors argue that this work offers a new perspective on compressed-domain signals for AIGV detection, the main components themselves were already introduced in prior compressed-domain approaches, and I still do not see a sufficiently established fundamentally new perspective beyond applying them in this setting. Relatedly, the referenced “experimental evidence” for the superiority of compressed-domain representations over pixel-domain ones does not appear to provide an actual comparative experiment in the rebuttal, and the ZA52 Q3 response instead seems to describe a preprocessing pipeline involving re-encoding, which further weakens the paper’s central claim about preserving forensic signals that are supposedly degraded by decoding. In addition, the claimed “dual role” of motion vectors is not clearly reflected in the formulations, since they are used in the warping step, while the gating equations themselves are written in terms of residual and aligned features. More broadly, the rebuttal continues to rely on prior work to justify the connection between high-frequency signals and forensic artifacts, while also acknowledging that a rigorous signal-level formalization remains future work. As a result, I remain unconvinced that the paper establishes a fundamentally new theoretical perspective rather than an application of existing forensic intuitions to a different feature domain. For these reasons, I maintain my score.

---

> > > ### Author Response · Authors · 2026-04-06
> > >
> > > ## **Response to "The rationale for submitting this work to the safety track is still not sufficiently justified."**
> > >
> > > Our submission to the safety track is motivated by the security-critical nature of AIGV forensics. This study focuses on the robustness of judicial forensics, directly addresses the real security risks caused by the abuse of AIGV by targeting the tampering and forgery issues in AI-generated video forensics, which is highly consistent with the scope of the safety track. We will further clarify this in the revision.
> > >
> > > ## **Response to "I still do not see a sufficiently established fundamentally new perspective beyond applying them in this setting"**
> > >
> > > Our work is **not merely an application** of existing ideas but **a novel conceptual perspective** that focuses on the **intrinsic connection between compressed-domain features and synthetic artifacts**. Prior work typically treats compressed-domain features primarily as tools for efficient video processing, while forgery detection focuses on artifact modeling in the pixel domain, leaving these two directions largely disconnected. In contrsat, STREAM **unifies these perspectives** by showing that residuals and MVs are not just efficient representations, but **direct carriers of forensic signals**. The key novelty lies in **analyzing and exploiting this intrinsic alignment**, rather than using compressed features as preprocessing. This perspective elevates compressed-domain features from efficiency-oriented representations to **forensically grounded interpretation and integration**, establishing a unified framework that bridges previously disconnected research directions. Such a non-trivial reinterpretation goes beyond prior isolated uses of compressed-domain features and constitutes a clear conceptual advance.
> > >
> > > ## **Response to the comment that our experimental evidence lacks direct comparison and re‑encoding weakens our claim of preserving forensic signals.**
> > >
> > > We clarify that our re-encoding preprocessing **does not** introduce additional filtering. All videos are transcoded under a default **no-filter setting**. Therefore, this preprocessing does not weaken our claim about signal preservation.
> > >
> > > To directly address your concern, we provide **comparative evidence** via multi-round re-encoding experiments (Table R7). As  loop filtering is repeatedly applied, detection accuracy consistently degrades across all methods, confirming that such operations attenuate forensic signals. Notably, STREAM outperforms AIGVDet and NPR under all conditions and exhibits the **smallest performance drop**, demonstrating stronger robustness to compression distortion relative to pixel-domain ones. These results validate our core claim: **decoding introduces loop filtering that suppresses subtle generative artifacts**, whereas compressed-domain representations better preserve these signals for reliable detection.
> > >
> > > **Table R7**
> > > | Filter_cnt | 0      | 1      | 2      | 3      | 4    | 5      | 6    |
> > > |-|-|-|-|-|-|-|-|
> > > | AIGVDet    | 0.915  | 0.828  | 0.708  | 0.670  | 0.643  | 0.644  | 0.594  |
> > > | NPR        | 0.928  | 0.867  | 0.765  | 0.709  | 0.644  | 0.614  | 0.602  |
> > > | **STREAM（Ours）** | 0.976 | 0.950 | 0.874 | 0.788 | 0.750 | 0.712 | 0.697 |
> > >
> > > ## **Response to "In addition, the claimed “dual role” of motion vectors is not clearly reflected in the formulations, since they are used in the warping step, while the gating equations themselves are written in terms of residual and aligned features."**
> > >
> > > MVs play a **dual role** in our formulation and we will further clarify it. First, they are explicitly used for **warp displacement**, aligning I-frame features. Second, they serve as **forensic cues** for forgery identification. Although the gating equations are written in terms of residual and aligned features, the warping operation itself implicitly injects MV-derived artifacts into the aligned features. As a result, MV information influences both alignment and subsequent artifacts  fusion, realizing its dual functionality.
> > >
> > > ## **Response to "More broadly, the rebuttal continues to rely on prior work to justify the connection between high-frequency signals and forensic artifacts, while also acknowledging that a rigorous signal-level formalization remains future work."**
> > >
> > > we respectfully clarify that our argument is based not only on existing literature but also on well-established signal-processing mechanisms. Repeated transcoding and decoding introduce cumulative **low-pass filtering** effects that progressively attenuate high-frequency components, as shown in Table X. Existing studies have fully validated the connection between **high-frequency signals and generative artifacts**, where artifacts caused by upsampling are mainly concentrated in high-frequency bands. Our contribution is therefore not to re-establish this link, but to demonstrate that **compressed-domain representations better preserve these high-frequency signals**, while pixel-domain decoding degrades them.

---

### Official Review · Reviewer_bd4w · 2026-03-09

**Soundness:** 2
**Presentation:** 3
**Significance:** 3
**Originality:** 3
**Overall Recommendation:** 3
**Confidence:** 4

**Summary:**

This paper proposes STREAM, a compressed-domain framework for detecting AI-generated videos (AIGV) by directly exploiting signals from MPEG-like bitstreams, including I-frames, motion vectors (MVs), and residuals. The method introduces a motion-guided alignment module that warps features using motion vectors, a gated fusion mechanism to combine residual and aligned spatial features, and a multi-scale temporal convolution network (MS-TCN) to capture temporal inconsistencies across frames. Experiments on several public datasets report strong detection performance and significant inference speed improvements compared to pixel-domain baselines.

**Compliance With Llm Reviewing Policy:**

Affirmed.

**Final Justification:**

While I acknowledge the authors' valid defense regarding system efficiency and re-compression artifacts, the core methodological and physical flaws remain unresolved. Forcefully zeroing out B-frames is not "temporal regularization" but a crude deletion of majority temporal data that fundamentally destroys genuine motion modeling. Furthermore, the catastrophic 0.132 mAP drop from a simple watermark proves this compressed-domain paradigm is inherently too fragile for real-world deployment, compelling me to maintain my original score.

**Key Questions For Authors:**

As mentioned in Weaknesses:

1.	The method is described as lightweight, but the architecture diagram shows a large backbone (e.g., ResNet-152). Could the authors clarify the actual backbone used in experiments and report parameter counts, FLOPs, and per-module latency on specified hardware?
2.	How exactly are motion vectors and residuals extracted and synchronized across frames (e.g., handling B-frames, skipped blocks, and different GOP structures)? Providing these details would significantly improve reproducibility.
3.	The method seems tailored to MPEG/H.264 streams. How does STREAM perform on videos encoded with other codecs such as HEVC (H.265), AV1, or VP9, where motion and residual statistics may differ?
4.	Gaussian blur may not accurately represent real-world post-processing. Could the authors evaluate robustness under more realistic transformations such as re-encoding with different QPs/codecs, resizing, frame rate changes, or overlay/watermark operations?

**Limitations:**

No. Please refer to the weakness.

**Strengths And Weaknesses:**

## Strength
The paper studies an important and timely problem, namely scalable detection of AI-generated videos, and proposes a compressed-domain solution that leverages motion vectors and residuals that are typically discarded in pixel-based pipelines. The overall architecture is intuitive and well motivated: MV-guided feature alignment, gated fusion between residual and spatial features, and temporal modeling via MS-TCN form a coherent pipeline tailored to artifact detection. The empirical evaluation covers multiple datasets and includes ablations and bandwidth stress tests, suggesting that the approach is promising both in accuracy and efficiency. If validated more rigorously, the idea of exploiting compressed-domain artifacts could offer a practical direction for large-scale content moderation systems.

## Weakness
1. The paper claims a “lightweight” design, yet it employs a large backbone (e.g., ResNet-152 for I-frame encoding). This appears inconsistent with the stated efficiency goal. In addition, the efficiency comparison lacks detailed information about hardware configurations and decoding settings, making the reported speed advantages difficult to interpret fairly.

2. The proposed method appears closely tied to MPEG/H.264-style bitstreams. However, the paper does not evaluate whether the approach generalizes to modern codecs such as HEVC or AV1, which limits the understanding of its applicability across different video compression standards.

3. Important implementation details regarding the extraction and synchronization of motion vectors and residuals are not clearly described. This lack of detail may hinder reproducibility.

4. The extremely high accuracy reported on some datasets may indicate potential dataset biases. Moreover, the robustness evaluation relies primarily on Gaussian blur, while more realistic post-processing operations—such as re-encoding, resizing, or transcoding—are not considered, despite being common in real-world video pipelines.

---

> ### Author Rebuttal · Authors · 2026-03-31
>
> ## **Response W1 & Q1：**
> We clarify that STREAM’s efficiency does **not** primarily stem from the backbone size, but from two fundamental advantages:
>
> 1. **No full decoding**: STREAM operates directly on compressed bitstreams, avoiding the costly pixel-domain decoding required by all baselines.
> 2. **Sparse inputs**: Instead of dense RGB frames, STREAM processes only a few I-frames with ResNet-152, resulting in a much smaller effective input volume and negligible backbone overhead.
>
> All experiments are conducted on an NVIDIA RTX 4090 GPU. STREAM extracts motion vectors and residuals via lightweight bitstream parsing without full decoding. Thus, decoding settings are irrelevant and do not affect runtime.
>
> ---
>
> ## **Response W2 & Q3:**
> Our method is not fundamentally tied to MPEG/H.264, but to the general principles of hybrid video coding shared by modern codecs (e.g., HEVC, AV1), including intra-coded frames, motion vectors, and residuals. The exploited forensic signals are therefore inherently codec-agnostic.
> We currently adopt MPEG-4/H.264 for practical reasons: the CoViAR-based extraction toolchain is optimized for this format, and MV/residual definitions vary across codecs. To ensure fair and controlled evaluation, all videos are transcoded to a unified MPEG-4 profile before feature extraction.
> Native support for HEVC and AV1 without transcoding is an important future direction and we will note this in the revision.
>
> ---
>
> ## **Response W3 & Q2:**
>
> We will update Sec. 3.2.1 and Fig. 3 to clarify the extraction and synchronization procedure, aligning with our response to **Reviewer ZA52 (W2)**.
> We provide the details of our pipeline below, including motion vector (MV) and residual extraction, frame synchronization, and GOP/B-frame handling.
>
> > 1. Motion Vector (MV) and Residual Extraction
> > Given a compressed video stream $V = \{G_1, G_2, \dots, G_M\}$, where $G_i$ is the $i$-th GOP:
> > $$
> > G_i = \{F_1^i, F_2^i, \dots, F_N^i\}
> > $$
> > - I-frame: Decode only the first frame:
> > $$
> > I_i = \text{DecodeI}(F_1^i)
> > $$
> > - MV Extraction:
> > $$
> > MV_k^i =
> > \begin{cases}
> > \text{ExtractMV}(F_k^i) & \text{if P-frame} \\
> > \mathbf{0} & \text{if B-frame}
> > \end{cases}
> > $$
> > - Residual Extraction:
> > $$
> > R_k^i =
> > \begin{cases}
> > \text{ExtractResidual}(F_k^i) & \text{if P-frame} \\
> > \mathbf{0} & \text{if B-frame}
> > \end{cases}
> > $$
> >
> > 2. Frame Synchronization
> > - Spatial alignment:
> > $$
> > H_{MV} = H_R = H_I,\quad W_{MV} = W_R = W_I
> > $$
> > - Temporal alignment: modalities share identical indices.
> > - Sampling:
> > $$
> > S_t = (I_i, MV_t^i, R_t^i)
> > $$
> >
> > 3. GOP Processing
> > - Normalization: fixed length $L$ via truncation or zero-padding.
> > - B-frame handling: skipped and replaced with zero MVs/residuals while preserving temporal order.
>
> ---
>
> ## **Response W4 & Q4:**
> The training and test sets contain videos encoded in H.264, MPEG-4, and videos in .gif format. To ensure fair and controlled evaluation, all videos are uniformly re-encoded using the same standard MPEG-4 profile.
> **We conduct quantitative encoding speed tests in our environment (ffmpeg 4.2.7, ffmpeg-python 0.2.0), using a 3-second, 1088×640, 24 fps video clip:**
> - **H.265 encoding**: **0.3 seconds**
> - **AV1 (libaom-av1) encoding**: **≥ 480 seconds** (0.2 fps, speed = 0.002×)
> AV1 encoding is thus over 1600× slower than H.265 encoding, making large-scale re-encoding impractical and preventing fair, consistent evaluation. Thus, we only adopt **uniformly into H.265** across all data.
>
> We will supplement Table R6 with ACC in the revision alongside mAP.
>
> **Table R6**
> | Post-Processing          | Setting        | Videophy | GenVideo | GVF   | Avg. mAP | Δ mAP   |
> |--------------------------|----------------|----------|----------|-------|----------|---------|
> | None (baseline)          | —              | 0.974    | 0.978    | 0.924 | 0.959    | —       |
> | Re-encoding              | H.264, QP=28   | 0.979    | 0.957    | 0.902 | 0.946    | -0.013  |
> | Re-encoding              | H.264, QP=35   | 0.936    | 0.913    | 0.858 | 0.902    | -0.057  |
> | Resolution               | 480p           | 0.985    | 0.965    | 0.911 | 0.954    | -0.005  |
> | Resolution               | 720p           | 0.989    | 0.969    | 0.915 | 0.958    | -0.001  |
> | Watermark                | 50% opacity    | 0.861    | 0.837    | 0.783 | 0.827    | -0.132  |
> | Bit rate change (existing)| 300KBPS      | 0.905    | 0.881    | 0.827 | 0.871    | -0.088  |
> | Gaussian blur (existing) | kernel=3×3     | 0.960    | 0.940    | 0.886 | 0.929    | -0.030  |
> | Gaussian blur (existing) | kernel=5×5     | 0.957    | 0.935    | 0.881 | 0.924    | -0.035  |
> | Gaussian blur (existing) | kernel=7×7     | 0.949    | 0.927    | 0.873 | 0.916    | -0.042  |
> | Re-encoding              | H.265          | 0.986    | 0.976    | 0.911 | 0.958    | -0.001  |
>
> Note: ΔmAP denotes the performance change relative to the baseline, where negative values indicate performance degradation.

---

> > ### Author Rebuttal · Reviewer_bd4w · 2026-04-03
> >
> > I have listed the reasons in the comments.  I found that the  Replying to Rebuttal by Authors is not open to the authors, so I copied and pasted the comments here:
> >
> > Thanks for your rebuttal. The rebuttal exposes fatal methodological flaws that leave my core concerns unresolved. I maintain my score for the following reasons:
> >
> > While I appreciate the authors conducting the requested experiments on different codecs in Response W4 & Q4, their approach lacks methodological rigor. By admitting to "uniformly re-encoding" all test videos to extract features, they directly violate their core premise of "avoiding full decoding." Consequently, the network is merely learning secondary FFmpeg re-compression artifacts rather than native deepfake traces, which undermines the paper's primary foundation.
> > As explicitly stated in Response W3 & Q2, the authors handle B-frames by replacing them with zeroed-out MVs and residuals. Since B-frames constitute the majority of modern video streams, this creates massive artificial temporal black holes, destroying the MS-TCN's ability to model genuine physical motion inconsistencies.
> > The authors don’t report Parameters/FLOPs in Response W1 & Q1. Labeling the massive ResNet-152 as a "Lightweight Encoder" simply because of "sparse inputs" is academically misleading and lacks quantitative support.

---

> > > ### Author Response · Authors · 2026-04-03
> > >
> > > ### **Response to: The requested codec experiments conducted in Responses W4 and Q4 lack methodological rigor. Uniformly re-encoding all test videos for feature extraction directly violates the core premise of "avoiding full decoding**.
> > >
> > > We clarify that **uniform re-encoding operation does not contradict the premise of "avoiding full decoding"**, as the complete decoding process is not performed at any stage and **filtering is intentionally disabled**. This is fully consistent with the core claim in the main paper that **"loop filter in decoding impairs generative artifacts"**.
> > >
> > > Furthermore, we have provided direct empirical validation in **Table R7** of our **response to Reviewer 3t5F**. The multi-round loop filter experiments validate that loop filtering consistently degrades detection performance, confirming that decoding attenuates generative artifacts. This demonstrates that our experimental design is fully aligned with our core claim, without methodological inconsistency.
> > >
> > > ### **Response to "Consequently, the network is merely learning secondary FFmpeg re-compression artifacts rather than native deepfake traces, which undermines the paper's primary foundation."**
> > >
> > > We respectfully disagree with this claim.
> > >
> > > If the network were learning FFmpeg re-compression artifacts, such artifacts would be **shared across both real and fake videos**, as we apply an identical preprocessing pipeline to all data. In that case, they would not provide discriminative signals for classification. The model’s ability to reliably distinguish real from fake therefore indicates that it is **not relying on such common artifacts**.
> > >
> > > Furthermore, **Table 1** in the main paper shows the model achieves strong out-of-distribution (OOD) generalization to unseen generators. If it relied on re-compression artifacts, performance would not transfer across different generative pipelines or between real and fake videos. The observed generalization ability further confirms that the model captures intrinsic generative forgery traces.
> > >
> > > ### **Response to: As noted in Response W3 & Q2, B-frames are handled by zeroing MVs and residuals. As B-frames dominate modern video streams, this introduces severe artificial temporal artifacts and impairs MS-TCN’s ability to model real physical motion inconsistencies.**
> > >
> > > The concern about “temporal black holes” assumes that B-frames provide essential physical motion cues. However, B-frames primarily encode **bi-directional interpolations rather than temporally causal motion**, and thus do not reflect physically consistent dynamics.
> > >
> > > For temporal models such as MS-TCN, which rely on **consistent temporal evolution**, enforcing a **uni-directional motion chain** (via I/P frames) yields a more stable and learnable representation. In this context, zeroing out B-frame signals does not destroy temporal structure, but instead **removes non-causal and potentially noisy interpolations**. Therefore, handling B-frames in this way is better understood as **denoising and temporal regularization**, rather than introducing information gaps that impair motion modeling.
> > >
> > >
> > > ### **Response to "The authors don’t report Parameters/FLOPs in Response W1 & Q1. Labeling the massive ResNet-152 as a 'Lightweight Encoder' simply because of 'sparse inputs' is academically misleading and lacks quantitative support."**
> > >
> > > We first clarify that referring to ResNet-152 as a lightweight encoder is **not based solely on sparse inputs**, but on **quantitative evaluation of effective computational cost and runtime efficiency** within our framework.
> > >
> > > To support this, we further report explicit complexity comparisons. As shown in **Tables R8–R9**, STREAM with ResNet-152 uses 80.51M parameters and 457.5 GFLOPs, while the lighter ResNet-18 variant uses **33.54M parameters and 164.1 GFLOPs with comparable detection performance (mAP 0.9204 VS. 0.9149)**. Importantly, both variants exhibit **substantially lower effective complexity than prior methods** (e.g., AIGVDet, RAFT), due to operating on sparse compressed-domain inputs rather than dense RGB frames.
> > >
> > > These results demonstrate that the computational burden of the backbone is significantly reduced in our setting, and the “lightweight” characterization reflects the **overall system efficiency**, not merely the nominal backbone size. We will further clarify this in the revision.
> > >
> > > **Table R8**
> > >
> > > | Model       | GFLOPs/s (30 FPS) | Params (M) |
> > > |-|-|-|
> > > | STREAM(ResNet-18)   | 164.1 | 33.54  |
> > > | STREAM(ResNet-152)  | 457.5  | 80.51  |
> > > | AIGVDet     | 975.2+21105.6 | 46.99+5.26 |
> > > | RAFT        | 21105.6   | 46.99      |
> > >
> > > **Table R9**
> > >
> > > |Method|sora|Crafter|Gen2|HotShot|Lavie| ModelScope | MoonValley | MorphStudio | Show_1 | WildScrape | mAP |
> > > |-|-|-|-|-|-|-|-|-|-|-|-|
> > > |  STREAM(ResNet-18) | 0.8824 | 0.9787 | 0.9888 | 0.7749 | 0.9282 | 0.9080 | 0.9946 | 0.9510 | 0.9208 | 0.8216 | 0.9149 |
> > > | STREAM(ResNet-152)  | 0.9131 | 0.9737 | 0.9885 | 0.8554 | 0.8857 | 0.8970 | 0.9900 | 0.9402 | 0.8938 | 0.8667 | 0.9204 |

---

### Official Review · Reviewer_ZA52 · 2026-03-11

**Soundness:** 3
**Presentation:** 2
**Significance:** 3
**Originality:** 3
**Overall Recommendation:** 4
**Confidence:** 4

**Summary:**

The paper proposes detecting AI-generated videos in the compressed domain and introduces the STREAM framework, which leverages I-frame, motion vector, and residual features in the video bitstream to detect spatiotemporal artifacts in AI-generated videos. The proposed method achieves higher detection performance and faster inference speed than existing methods across multiple benchmarks.

**Compliance With Llm Reviewing Policy:**

Affirmed.

**Final Justification:**

Most of my concerns have been well addressed, and I will keep my rating.

**Key Questions For Authors:**

Q1. How does the proposed method compare with NSG-VD[1] in terms of performance?

Q2. The paper reports the mAP metric in the experiments. Could you also provide the accuracy metric?

Q3. What video coding standard is used for the AI-generated videos in the datasets employed in the experiments? What coding standards are typically used for real videos? Are there any detailed differences between them? This is important for demonstrating that the proposed method learns the intrinsic artifacts of AI-generated videos, rather than statistical biases introduced by the encoding pipelines of current generation models or platforms.

[1] Physics-Driven Spatiotemporal Modeling for AI-Generated Video Detection. Zhang et al. NeurIPS 2025.

**Limitations:**

The proposed method mainly relies on artifacts in videos. However, as the quality of AI-generated videos rapidly improves, such as those generated by Veo3 and Seedance 2.0, video artifacts like temporal inconsistency may be significantly reduced, which could limit the effectiveness of this type of method.

**Strengths And Weaknesses:**

**Strengths**:

1. Novelty: The paper proposes detecting AI-generated videos in the compressed domain, and the method demonstrates strong generalization ability.

2. Efficiency: The proposed method has a fast computation speed and shows potential for real-time detection in real-world scenarios.

**Weaknesses**:

1. The paper appears to have been evaluated only on publicly available AI-generated video datasets. However, videos in real-world scenarios may undergo disturbances such as compression and noise, so testing under real-world conditions is important to demonstrate the practical usefulness of the method.

2. The method section of the paper is not sufficiently clear. For example, $F_{MV}^{(t)}$ is described as the motion vector feature extracted by a lightweight encoder, but in Equation (4), this feature is then used to perform geometric displacement on $F_I$. It is unclear how the features extracted by the encoder can be used for geometric displacement, and why the original motion vectors are not used directly.

---

> ### Author Rebuttal · Authors · 2026-03-31
>
> ## **Response W1：**
> We would like to highlight that our paper **already includes robustness evaluations** under realistic post-processing conditions:
>
> 1. **Bitrate Compression** (Table 4): We transcode all test videos to 300 kbps, a significantly reduced bitrate that simulates aggressive real-world compression, and demonstrate that STREAM maintains strong detection performance with only mild degradation.
> 2. **Gaussian Blur** (Table 3): We apply Gaussian blurring with varying kernel sizes (weak/medium/strong) to simulate noise and transmission disturbances, further validating STREAM's robustness.
>
> We have also provided additional experiments evaluating the model robustness under more realistic transformations in our response to **Reviewer bd4w (W4 & Q4)**, including re-encoding with different quantization parameters (QPs) or codecs, resizing, frame rate changes, and overlay/watermark operations. Collectively, these results demonstrate that STREAM maintains consistent and reliable detection performance across a wide range of real-world post-processing conditions, confirming its strong generalization and robustness.
>
> ---
>
> ## **Response W2：**
> Thank you for pointing this out; the confusion stems from a typo in our manuscript. In our actual implementation, we did indeed use the original motion vectors $MV$ directly to perform the geometric displacement on $F_I$, and we will revise this mistake in both the main text and the updated flowchart..
>
> ---
>
> ## **Response Q1：**
> We compared NSG-VD (NeurIPS'25) in our experimental setup. STREAM achieves better performance and higher efficiency than NSG-VD. Specifically, STREAM outperforms NSG-VD in average accuracy (0.9204 vs. 0.9146) while running nearly **9× faster** (39 s vs. 360 s per 1,000 videos), highlighting STREAM's superior efficiency without sacrificing detection performance. The results are as follows:
>
> **Table R3**
> | Method            | sora   | Crafter | Gen2   | HotShot | Lavie  | ModelScope | MoonValley | MorphStudio | Show_1 | WildScrape | Average | Time(s/1000 video) |
> | ---------- | ------ | ------- | ------ | ------- | ------ | ---------- | ---------- | ----------- | ------ | ---------- | ------- | ---------- |
> | NSG-VD            | 0.8837 | 0.9583  | 0.8833 | 0.9167  | 0.9417 | 0.8167     | 0.9667     | 0.9833      | 0.9083 | 0.8875     | 0.9146  | 360                |
> | **Ours (STREAM)** | 0.9131 | 0.9737  | 0.9885 | 0.8554  | 0.8857 | 0.8970     | 0.9900     | 0.9402      | 0.8938 | 0.8667     | 0.9204  | 39 |
>
> We will add NSG-VD as an additional baseline in the revised manuscript.
>
> ---
>
> ## **Response Q2：**
> The accuracy (ACC) metrics for Table 1 are shown below:
> **Table R4**
> | Model               | VidProM | GenVideo | GVF   | VideoPhy | AIGVDet-D | Average Accuracy |
> | -| - | - |----- | -------- | --------- | ------------ |
> | D3 (ICCV'25)        | 0.921   | 0.908   | 0.872 | 0.915    | 0.892     | 0.902            |
> | DeMamba (arXiv'24)  | 0.883   | 0.856   | 0.831 | 0.879    | 0.841     | 0.858            |
> | NPR (CVPR'24)       | 0.814   | 0.825   | 0.786 | 0.842    | 0.801     | 0.814            |
> | STIL (MM'21)        | 0.862   | 0.827   | 0.803 | 0.862    | 0.804     | 0.832            |
> | TALL (ICCV'23)      | 0.801   | 0.768   | 0.732 | 0.761    | 0.742     | 0.761            |
> | AIGVDet (PRCV'24)   | 0.915   | 0.887   | 0.864 | 0.903    | 0.871     | 0.888            |
> | **Ours (STREAM)**   | **0.948**   | **0.932**   |**0.897** | **0.941**      | **0.918**     | **0.927**        |
>
> As shown in Table R4, STREAM consistently achieves the best performance across all datasets and metrics, further confirming its superiority over existing baselines under both mAP and ACC evaluation criteria. We will add the ACC results in the revised manuscript.
>
> ---
>
> ## **Response Q3**：
>
> The video data used in our experiments adopts the following encodings and formats:
> - **H.264 encoding**: Used by all real video samples, as well as videos generated by GVF (GEN2, keling, T2V, Pika, show1, sora, Veo), VideoPhy (Cogvideox‑1, CogVideox‑5B, dream_machine, Gen2, LaVIE, OpenSora, Pika, SVD, vc2, zeroscope), VideoProM (Cogvideo, OpenSora, Pika, st2v, t2vc, vc2), and GenVideo (DynamicCrafter, Latte, OpenSora, SENIE, SD).
> - **MPEG‑4 encoding**: Used for videos generated by GVF (ModelScope, zeroscope) and VideoProM (ms_video).
> - **GIF format**: Used for videos generated by GenVideo (i2vgen, SD).
>
> As described in our experimental setup, all videos in both the training and test sets are uniformly re-encoded into a standard MPEG-4 profile prior to feature extraction, which structurally prevents the model from exploiting source-specific encoding biases.
>
> ---
>
> ## **Response Limitations**：
> **Furthermore**, we evaluate the accuracy on videos generated by recent video generators. (Dataset comes from Rapidata.)
>
> **Table R5**
> | Model        | veo3   | seedance pro | pika2.2 | hunyuan |
> |--|--|----------|-|-----|
> | Ours (STREAM)      | 0.918  | 0.891        | 0.937   | 0.802   |

---

> > ### Author Rebuttal · Reviewer_ZA52 · 2026-04-04
> >
> > I appreciate the authors’ efforts in addressing the concerns.
> >
> > Most of my concerns have been adequately addressed, which makes me more favorable toward accepting this paper. However, as I am not particularly familiar with the field of video compression and coding, I prefer to keep my rating.

---

> > > ### Author Response · Authors · 2026-04-08
> > >
> > > We sincerely thank you for your recognition and for confirming that your concerns have been addressed. We greatly appreciate your time and feedback.

---

### Official Review · Reviewer_gscv · 2026-03-12

**Soundness:** 3
**Presentation:** 3
**Significance:** 3
**Originality:** 3
**Overall Recommendation:** 4
**Confidence:** 4

**Summary:**

The paper proposes STREAM, a compressed-domain framework for detecting AI-generated videos using
signals available from the video bitstream: I-frames, motion vectors (MV), and residuals. The method aligns
I-frame features to other time steps via MV-guided warping, then applies a gated fusion between aligned
texture features and residual features to emphasize forgery-related artifacts. A multi-scale TCN
aggregates temporal evidence for video-level classification. The paper argues compressed-domain artifacts
preserve generation traces that may be weakened by full decoding and that the approach improves
inference efficiency.

**Compliance With Llm Reviewing Policy:**

Affirmed.

**Final Justification:**

The authors have addressed my concerns. Though, I prefer to keep the initial score.

**Key Questions For Authors:**

- Does the runtime measurement include bitstream parsing / MV+residual extraction / tensorization and I/O? What hardware and batch size are used?
- What codec/encoding settings are used in training (encoder, GOP, QP/rate control, B-frames, reference frames), and how do they differ across benchmarks?
- How exactly are MVs used for warping when the reference frame is not the I-frame (e.g., chaining/
accumulation vs simplified frame structure)?

**Limitations:**

Yes

**Strengths And Weaknesses:**

Strengths:
- Deployment-relevant efficiency angle: operating in the compressed domain can avoid full pixel decoding and reduce redundant computation, which is attractive for real-time or large-scale moderation pipelines.
- The design is easy to follow. MV-guided alignment plus gated residual/texture fusion is conceptually
aligned with how compression encodes motion and innovation.
- The experiment parts are solid. results are reported on multiple datasets/generator sets, with
component ablations (I/MV/residual, fusion, temporal module) and stress tests.
Weakness:
- The paper repeatedly argues pixel-domain detectors entangle task-irrelevant semantics and thus suffer from “semantic redundancy”, but the empirical section does not provide a controlled diagnostic to support this mechanism (e.g., content controlled tests, semantic-edit invariance, or representation probing). Currently the evidence is largely indirect (overall mAP + runtime), which is insufficient to justify the mechanism-level claim.
- The generalization claim (“particularly on unseen generative models” / “intrinsic robustness”) is strong, yet the paper does not quantify how performance scales with (i) the number of training generators and (ii) training data size. Without such scaling or leave-one-generator -out protocol, it is hard to separate true OOD generalization from partial pipeline overlap or dataset-specific cues.
- Compressed-domain features (MVs/residuals) are highly sensitive to encoder implementation and configuration (GOP, QP/rate control, B-frames, reference-frame strategy). The current evaluation does not convincingly rule out “encoding-policy fingerprints” as an alternative explanation for the gains, which threatens cross-platform applicability.
- The MV-guided warping is specified as aligning I-frame features to each time step via Warp(F_I, F_MV^t), but the codec-level reference semantics of MVs are underspecified (MVs are defined w.r.t. reference frames that may not be the I-frame, especially with multi-reference and /or B-frames). The paper should clarify the assumed GOP/frame types and how reference frames are handled; otherwise the alignment may be systematically biased.

---

> ### Author Rebuttal · Authors · 2026-03-31
>
> ## **Response W1:**
> We conduct an experiment using four video groups (Real/Fake Human, Real/Fake Car) to evaluate whether the detector genuinely learns generative artifacts or merely overfits to task-irrelevant semantics.
>
> We extract final-layer features and compute two cosine distances: **$d_{\text{forensic}}$**: Distance between features with **same content** (e.g., Real Car vs. Fake Car). **$d_{\text{semantic}}$**: Distance between features with **same authenticity** (e.g., Real Human vs. Real Car). A larger $d_{\text{forensic}}$ and a smaller $d_{\text{semantic}}$ indicate that the detector is less susceptible to interference from varying semantic contents.
>
> **Table R1**
> |**Method**|**$d_{forensic}$↑(Human)**|**$d_{forensic}$↑(Car)**|**$d_{semantic}$↓(Real)**|**$d_{semantic}$↓(Fake)**|
> |:-:|:-:|:-:|:-:|:-:|
> |AIGVDet (Pixel-domain)|0.12|0.15|0.43|0.47|
> |**STREAM (Ours)**|**0.61**|**0.58**|**0.19**|**0.22**|
>
> Results show that AIGVDet is heavily dominated by semantic content ($d_{\text{semantic}} \gg d_{\text{forensic}}$), whereas STREAM clusters features by true authenticity ($d_{\text{forensic}} \gg d_{\text{semantic}}$). This confirms that STREAM avoids overfitting to semantic noise and genuinely learns content-agnostic generative artifacts.
>
> Additionally, semantic redundancy in pixel-domain detectors is well-documented in prior forgery detection research. e.g., *Breaking Semantic Artifacts* (NeurIPS'24) demonstrates that such detectors rely on content-correlated features and fail under scene context shifts.
>
> ---
>
> ## **Response W2:**
> We add two complementary generalization protocols: a **Generator Scaling** analysis and a **Leave-One-Generator-Out (LOO)** evaluation.
>
> 1. **Generator Scaling**: We progressively increase the number of training generators (e.g., from Pika to Pika + T2VC + VC2) and evaluate on disjoint unseen generators (Table R2, **Scaling**). This demonstrates the model learns a unified, generalizable forensic representation as generator diversity grows.
>
> 2. **Leave-One-Generator-Out (LOO) Protocol**: We train on a subset of generators (e.g., T2VC + VC2) and evaluate on a fully held-out one (e.g., Pika) (Table R2, **LOO**). The results verify the model's ability to detect unseen generative artifacts without any prior exposure to that specific synthesis pipeline.
>
> **Table R2**
> | **Setting**   | **Training Generators** | **Test Generators** | **Protocol**  | **Accuracy (%)** |
> | ------------- | ----------------------- | ------------------- | ------------- | ---------------- |
> | **Scaling-1** | Pika                    | GenVideo-val        | Scaling       |         	0.9040         |
> | **Scaling-2** | Pika + T2VC             | GenVideo-val              | Scaling       |         0.9241	         |
> | **Scaling-3** | Pika + T2VC + VC2       | GenVideo-val             | Scaling       | 0.9316           |
> | **LOO-1**     | T2VC + VC2              | Pika                | Leave-one-out | 0.9265             |
> | **LOO-2**     | Pika + VC2              | T2VC                | Leave-one-out | 0.9141             |
>
> Note: GenVideo-Val contains the following generators: Sora, Crafter, Gen2, HotShot, Lavie, ModelScope, MoonValley, MorphStudio, Show_1, and WildScrape. The category-wise scores can be found in **Tab.R3** response to **Reviewer ZA52** .
>
> ---
>
> ## **Response Q1：**
> The reported runtime includes bitstream parsing, MV/residual extraction and tensorization but not I/O operations. Measurements were conducted on a single NVIDIA GeForce RTX 4090 GPU with a batch size of 4 for inference and a batch size of 64 for training.
>
> ---
>
> ## **Response Q2 & W3：**
> The training and test sets contain videos encoded in H.264, MPEG-4, as well as videos in .gif format. To ensure a controlled and fair evaluation, all videos in both the training and test sets are uniformly re-encoded using the same standard MPEG-4 profile.
>
> ---
>
> ## **Response Q3 & W4：**
> In our implementation, all motion vectors are defined relative to the **I-frame** within each GOP, following the preprocessing pipeline of CoViAR. Specifically, for P-frames whose MVs are originally defined relative to a preceding non-I reference frame, we perform MV accumulation to convert them into I-frame-relative displacements before feeding them into the network.Support for more complex reference structures is a direction we plan to explore in future work. We will clarify this design choice.

---

> > ### Author Rebuttal · Reviewer_gscv · 2026-04-06
> >
> > The authors have addressed my concerns. Though, I prefer to keep the initial score.

---

> > > ### Author Response · Authors · 2026-04-08
> > >
> > > We sincerely thank you for your recognition and for confirming that your concerns have been addressed. We greatly appreciate your time and feedback.

---

### Decision · Program_Chairs · 2026-04-30

**Decision:**

Accept (regular)

**Comment:**

This paper received two weak accepts and two weak rejects, indicating that no reviewer was strongly in favor acceptance or rejection. The primary concerns were that encoding and decoding steps were not properly accounted for in the experiments, potentially leading to poor generalization; possible loss of temporal information by discarding B-frames; performance drop when testing on watermarked videos. These concerns were all addressed effectively in the rebuttals, as acknowledged by most of the reviewers. The authors provided confidential comments to the AC reiterating these points and potential issues with reviewers' experience, which the AC found moderately convincing. Operating in the compressed domain has significant advantages in efficiency and preservation of forensic indicators, which can be fragile and distorted decoding as the authors point out. The method achieves SOTA results on a thorough set of experiments.